# Multilingual estimation of political-party positioning:
# From label aggregation to long-input Transformers

**Dmitry Nikolaev**    **Tanise Ceron**    **Sebastian Padó**
Institute for Natural Language Processing, University of Stuttgart
dnikolaev@fastmail.com   {tanise.ceron,pado}@ims.uni-stuttgart.de

## Abstract

Scaling analysis is a technique in computational political science that assigns a political actor (e.g. politician or party) a score on a predefined scale based on a (typically long) body of text (e.g. a parliamentary speech or an election manifesto). For example, political scientists have often used the left–right scale to systematically analyse political landscapes of different countries. NLP methods for automatic scaling analysis can find broad application provided they (i) are able to deal with long texts and (ii) work robustly across domains and languages. In this work, we implement and compare two approaches to automatic scaling analysis of political-party manifestos: label aggregation, a pipeline strategy relying on annotations of individual statements from the manifestos, and long-input-Transformer-based models, which compute scaling values directly from raw text. We carry out the analysis of the Comparative Manifestos Project dataset across 41 countries and 27 languages and find that the task can be efficiently solved by state-of-the-art models, with label aggregation producing the best results.

## 1 Introduction

A widely used tool in computational political science is the so-called 'scaling analysis': a set of methods for representing political platforms as numbers on a certain scale, such as left–right, authoritarian–libertarian, or conservative–progressive (Laver et al., 2003; Slapin and Proksch, 2008; Diermeier et al., 2012; Lauderdale and Clark, 2014; Barberá, 2015). A wide variety of scales have been proposed in the literature, some based on political-theoretic considerations (Jahn, 2011), others more data-driven (Gabel and Huber, 2000; Albright, 2010; Rheault and Cochrane, 2020).

One well-established scoring scheme of this kind is the Standard Right–Left Scale, also known as the RILE score (Budge, 2013; Volkens et al., 2013).

It was developed in the framework of the Manifesto Research on Political Representation project (MARPOR), formerly known as the Comparative Manifestos Project (CMP),[1] which collects, annotates, and makes available a large collection of party platforms from different countries. The RILE score is a deductive, first-principle-based method for describing party positions geared towards the widest possible applicability across time and countries (Budge, 2013). For this very reason, it is rather conservative and inflexible and has been repeatedly criticised (see, e.g., Flentje et al., 2017). Despite this, it is widely used in computational political science for model validation (Rheault and Cochrane, 2020), as a dependent variable in regression analyses (Greene and O'Brien, 2016), or as a basis for party-stance analysis (Däubler and Benoit, 2022).

A major practical drawback of the RILE score is the fact that it is computed based on the labels manually assigned by MARPOR annotators to all statements in party manifestos (see Section 2 for details). This procedure is expensive and time consuming, which raises the question of whether we can adequately approximate the RILE score using natural-language-processing methods, especially in a multilingual setting. This will make it possible to efficiently perform analyses of political texts that have not yet been covered by the MARPOR project due to timing constraints (e.g. manifestos from upcoming elections), accidental gaps (e.g., Indonesia and the Philippines are not part of the dataset, and coverage of many countries, such as South Africa, is incomplete), or lack of resources (there are very few annotated manifestos from before 2000).

This work is a first step in this direction. Our contributions are the following:

1. Previous works on computational analysis of party positioning targeted a limited number of texts from a single country or several coun-

---

[1] https://manifestoproject.wzb.eu/

tries. We scale the analysis up to 41 countries and 27 languages, including comparatively low-resource languages (such as Georgian and Armenian) that have not been tackled before.

2. We contrast the label-aggregation approach, based on a statement-level classifier mimicking the work of a human annotator, with using long-input Transformer models predicting the scores directly from raw manifesto texts.

3. In the label-aggregation setting, we further compare the performance of multilingual-modelling-based and machine-translation-based approaches. While the former is more straightforward in the sense that a single base model can be directly used without any pre-processing, MT systems are easier to train for less-resource-rich languages, and only a single-language classifier is needed for predictions.

4. We evaluate the generalisability of models regarding two dimensions: local (moving to new countries) and temporal (moving from the past to the future). These correspond to different real-life research scenarios. We show that our methods deal reasonably with both cases.

The paper is structured as follows: In § 2, we provide more information on the MARPOR annotations and on how the RILE score is computed. The exact problem statement, different operationalization strategies, and the experimental setup are presented in § 3, while the results of the study are given in § 4. Additional discussion is provided in § 5. Section 6 surveys related work. Section 7 concludes the paper and discusses directions for future research.

## 2 MARPOR categories and political scales

**Categories** The annotations of the manifesto created in the framework of the Comparative Manifestos Project follow the project codebook (Volkens et al., 2020). Each statement of a given manifesto is annotated with a category representing a specific policy domain (e.g. *Military* or *Sustainability*). These categories can be identified via their names and numbers (e.g., 103, *Anti-Imperialism*).[2]

A key feature of MARPOR categories is that they are not stance-neutral. Thus, category 201,

---

[2]See Appendix B for the major categories with numbers.

*Freedom and Human Rights*, or subtypes thereof, are assigned to 'favourable mentions of importance of personal freedom and civil rights' (Volkens et al., 2020, 12). Some categories form binary oppositions (e.g. *Constitutionalism: Positive* vs. *Constitutionalism: Negative*), and some are purely one-sided (e.g. *Freedom* and *Democracy* have positive loadings and do not have negative counterparts). As a result, it is possible to derive inferences about political stances of different parties from category counts alone. This provides a straightforward operationalization of the political-science notion of *issue salience*, which is commonly used to analyse political positioning (Epstein and Segal, 2000) – the number of occurrences of a category correlates with how important it is for a party.

In total, there are 143 different categories, with 56 major categories, 32 sub-categories of the major categories, 54 additional categories, and the residual category 0.[3]

**Right–Left scale** A prominent way of analysing party positioning is the Standard Right–Left Scale, a.k.a. RILE score (Budge, 2013; Volkens et al., 2013). Originally developed in the framework of the MARPOR project, it has been consistently used in its publications and remains a standard reference scale for party positioning, despite a number of proposals to improve or replace it, both using theory-based and data-driven approaches (cf. Cochrane, 2015; Mölder, 2016; Flentje et al., 2017).

$$\text{RILE} = \frac{R - L}{R + L + O} \quad (1)$$

Eq. 1 shows the formula for computing the RILE score based on sets of categories defined by MARPOR as right-wing and left-wing, respectively. The categories belonging to the right and left sets are shown in Table 1.

$R$(right) and $L$(eft) are the percentages of statements labelled with categories from the two sets, and $O$ the percentage of other statements. The range of RILE is $[-1, 1]$. Large absolute values indicate extreme left and right programs, and values close to zero correspond to centrist manifestos with a balanced program.[4]

---

[3]In some manifestos, special label 'H' is attached to headings. As we cannot reliably automatically identify headings in new texts, H's were converted to 0's throughout.

[4]They could also arise from political programs where most statements are associated with neither left nor right, but such programs are rare in practice.

| | |
|---|---|
| Right emphasis | Military: Positive, Freedom, Human Rights, Constitutionalism: Positive, Political Authority, Free Enterprise, Economic Incentives, Protectionism: Negative, Economic Orthodoxy, Social Services Limitation, National Way of Life: Positive, Traditional Morality: Positive, Law and Order, Social Harmony |
| Left emphasis | Decolonisation, Anti-imperialism, Military: Negative, Peace, Internationalism: Positive, Democracy, Regulate Capitalism, Market, Economic Planning, Protectionism: Positive, Controlled Economy, Nationalisation, Social Services: Expansion, Education: Expansion, Labour Groups: Positive |

Table 1: The MARPOR categories used for calculating the RILE score.

## 3 Methods

### 3.1 Operationalization

**Label aggregation** As outlined above, we aim at automatically estimating positions of political parties on the Left–Right scale. An approach that closely mirrors the traditional MARPOR procedure would be to automatically label the sentences in the manifestos with MARPOR categories and aggregate them according to Eq. 1. Unfortunately, classifying the sentences is difficult, as we will show below. Reasons include the large number of labels, their uneven distribution, and the country-specific nature of manifestos.

However, the predicted categories arguably do not have to perfect – it may be sufficient for high-quality scaling analysis if the mistakes are uncorrelated so that, for example, the number of sentences mistakenly classified as left-leaning or neutral is close to the number of sentences mistakenly classified as right-leaning or neutral. One way to further raise the signal-to-noise ratio is to predict more high-level labels. To compute the RILE score, we do not require specific categories, but only a three-way classification (R[ight], L[eft], O[ther]), which is much more tractable. This approach can be easily mapped into other dimensions as long as there is a list of categories from MARPOR belonging to both poles of the scale.

**Direct prediction** As an alternative, we can define a function $\mathcal{T} \rightarrow [-1, 1]$ that directly maps a text to its RILE score, and approximate it with a neural regression model. Until recently, such an approach was infeasible due to the restrictions on the input length in the state-of-the-art embedding models: 512 or 1024 tokens depending on the model size, which is not enough to analyse longer texts. However, a new generation of long-input Transformers (LITs) based on lightweight variants of the self-attention mechanism increased the input limit to 4096 tokens or more (Tay et al., 2021). This still does not give us a way to compute a score for a whole text, but averages of RILE scores for 4095-token chunks of manifestos nearly perfectly correlate with gold manifesto-level scores (Spearman's $r > 0.99$), which makes by-chunk estimation a good proxy.

An additional motivation to pursue this avenue is provided by the fact that it not only removes the need to classify the labels of individual statements but also saves researchers the effort to identify statements in the first place. This is a non-trivial problem as, according to the MARPOR codebook, any sequence of words with a distinct meaning can be considered a statement. E.g., a sentence *All well-meaning citizens should strive to maintain the world peace* can be construed as a single example of the category *Peace*, or *all well-meaning citizens* can be assigned its own label of *Civic minded-ness*. In line with previous work, our aggregation-based approach assumes that statement boundaries are known, but in practice they will have to be predicted together with the labels, or the coding scheme must be simplified, e.g. by assigning a single 'majority' label to each sentence. By virtue of working with raw text spans, LITs do not have to make such compromises.

### 3.2 Problem settings

We consider two settings, corresponding to two different research scenarios. In the LEAVE-ONE-COUNTRY-OUT (X-COUNTRY) setting, we train the model on all data from $n - 1$ countries (split into training and development sets), and evaluate it one held-out country. This corresponds to the situation when manifestos from a country not yet covered by the MARPOR project, such as Indonesia, need to be analysed. This is repeated for all countries.

In the OLD-VS.-NEW (X-TIME) setting, we train the model on all data from before 2019 and evaluate it on the data from 2019–2021. This corresponds to the situation when new data from an already covered country become available.[5]

---

[5]Another application for this setting is the analysis of man-

## 3.3 Dataset

We use the annotated subset of the latest release of the MARPOR dataset (version 2022a; Lehmann et al., 2022a) augmented with the separately curated South American dataset (Lehmann et al., 2022b).[6] We excluded manifestos annotated before the year 2000 to obtain a more uniform training dataset. Furthermore, to ensure comparability between two approaches to cross-lingual modelling – preprocessing using machine translation and using a multilingual encoder (see § 3.4 below) – we excluded languages for which no pretrained free NMT system was readily available. This leaves us with 1314 manifestos from 41 different countries in 27 different languages.

In the X-COUNTRY setting, the rolling test set includes all of the data, while in the X-TIME setting it is much smaller (163,714 vs. 1,062,302 statements in the training set, i.e. around 13%) and has a weaker geographical coverage: only 18 countries have manifestos from 2019 and later.

The data for LITs have the same train-test general splits, but sentences in them were consecutively concatenated into text chunks of size no more than 4095 tokens (see Section 3.1), with a RILE score computed for each chunk based on its gold MARPOR labels. Chunks of size less than 1000 tokens were discarded.[7]

## 3.4 Models

The MARPOR dataset is multilingual, which raises the challenge of language transfer. The two current approaches in this case are using a multilingual encoder or machine translating all the data into the pivot language, usually English (Litschko et al., 2022; Srinivasan and Choi, 2022).

**Label aggregation** Here we experiment with both options. For the MULTILINGUAL-ENCODER TRACK (XLM-ENC), we extract the representation of the CLS token from XLM-RoBERTa base (in the X-COUNTRY setting) and XLM-RoBERTa base and large (in the X-TIME setting).[8] Throughout,

the classification head is a 2-layer MLP with the inner dimension of 1024 and tanh activation after the first layer.

In the X-COUNTRY setting, the model was then repeatedly trained for two epochs using cross-entropy loss and the AdamW optimiser (Loshchilov and Hutter, 2019) with the learning rate of $10^{-5}$.[9] In the X-TIME setting, the general setup is the same but the model was trained for five epochs with a checkpoint selected based on the dev-set accuracy.

For the MACHINE-TRANSLATION TRACK (MT), all manifestos are translated into English, for which the best MT systems and arguably the best pretrained encoders are available. The current MT systems, however, are still rather noisy, especially for non-WEIRD (Henrich et al., 2010) languages, which offsets the benefits of a stronger base model.

We use the EasyNMT toolkit[10] giving access to the Opus-MT models (Tiedemann and Thottingal, 2020). A cursory inspection of the translated sentences shows that the translation quality does vary across languages. However, even for manifestos whose source languages are difficult to translate (e.g. Georgian) the results produced by the classifier are still acceptable.

The translated sentences are encoded using pooled representations from `all-mpnet-base-v2`, a version of MPNet (Song et al., 2020) fine-tuned following the SBERT methodology (Reimers and Gurevych, 2019) and available on HuggingFace.[11] The same classification head was then used as in the XLM-ENC approach, as well as the same training parameters.

For each model, we aggregate the labels across manifesto sentences and compute its RILE score according to Eq. 1.

**Direct prediction** We experiment with two long-input encoder models: Longformer (Beltagy et al., 2020) and BigBird (Zaheer et al., 2020).[12] They are only available for English, and we apply them to the translated dataset. We use the embedding of

---

ifestos of smaller parties that did not win any seats in previous elections and were not included in the dataset. The converse – NEW-VS.-OLD – would permit running a historical analysis of party positioning within a country. We have not addressed this scenario due to the scarcity of annotations from before 2000.

[6]All data are available on the project web page: https://manifestoproject.wzb.eu/datasets.

[7]Statistics of the datasets are shown in Tables 8 (by country) and 9 (by language) in Appendix C.

[8]The necessity to train 41 different models on the full dataset in the X-COUNTRY setting made it impractical to use

the large model.

[9]The code for training and evaluating the models can be found at https://github.com/macleginn/party-positioning-code

[10]https://github.com/UKPLab/EasyNMT

[11]https://huggingface.co/sentence-transformers/all-mpnet-base-v2 Preliminary experiments showed, in agreement with the results of Ceron et al. (2022), that it consistently outperforms RoBERTa in monolingual settings.

[12]Pretrained models were downloaded from HuggingFace: https://huggingface.co/allenai/longformer-base-4096 and https://huggingface.co/google/bigbird-roberta-base.

the last layer's CLS token as input to a regression head. In the training step, each chunk receives a gold label computed from its sentences using Eq. 1. The final RILE score of each manifesto is the average of regression values of its chunks. The regression head is similar to the classification head described above with the final softmax layer replaced with a single node with tanh activation mapping the output into the $[-1, 1]$ range. The systems are trained using MSE loss.

### 3.5 From regression to classification with LITs

A possible concern about the direct computation of RILE scores, as we frame the task for LITs, is that the models may fail to implicitly recreate the labelling-and-aggregation pipeline and instead learn spurious shortcuts by observing correlations between properties of texts and their RILE scores, which will then hurt test performance.

To address this concern, we carry out an additional experiment where we make the models' task more comparable to what a human political analyst would do. We train the LITs in a binned-regression setting: the range of RILE scores is split into five regions, corresponding to *hard left* $[-1, -0.6)$, *centre left* $[-0.6, -0.2)$, *centrist* $[-0.2, 0.2)$, *centre right* $[0.2, 0.6)$, and *hard right* $[0.6, 1]$. The models are then trained to predict these classes instead of real-valued RILE scores using cross-entropy loss.

### 3.6 Evaluation metrics

For the label-aggregation models, we first diagnose the performance of the label classifiers using the weighted macro-averaged F1 score.

We then evaluate both the label-aggregation and the direct-prediction models on the target task of predicting RILE score. We use Spearman's correlation coefficient, which shows if our scores are monotonically related to those computed from gold annotations using Eq. 1. Additionally, we look at absolute values of errors and their directionality.

We evaluate the performance of the LIT-based classifiers in the binned-regression setting using accuracy and F1 score.

## 4 Results

The main results of the experiments are summarised in Tables 2 and 3. Sections 4.1 and 4.2 discuss the results while § 4.3 provides some detail about the strengths and weaknesses of the models.

|  |  | X-COUNTRY | | | X-TIME | | |
|---|---|---|---|---|---|---|---|
|  |  | XLM | MT | MAJ | XLM | MT | MAJ |
| CMP | Acc | 0.46 | 0.47 | 0.10 | 0.54 | 0.48 | 0.10 |
|  | F1 | 0.44 | 0.44 | 0.02 | 0.55 | 0.48 | 0.02 |
| RILE | Acc | 0.70 | 0.71 | 0.59 | 0.77 | 0.74 | 0.63 |
|  | F1 | 0.70 | 0.70 | 0.44 | 0.77 | 0.74 | 0.49 |

Table 2: The accuracies and class-weighted F1 scores of predicting all 143 MARPOR/CMP categories and 3 RILE-specific categories (*left*, *right*, *other*) in the leave-one-country-out (X-COUNTRY) and old-vs.-new (X-TIME) settings using a multilingual encoder (XLM-ENC) or preprocessing via machine translation (MT). MAJ is the majority-class baseline for each setting.

|  |  | RILE (CMP) | RILE (3-way) |
|---|---|---|---|
| X-COUNTRY | XLM | **0.73** | 0.72 |
|  | MT | 0.71 | 0.72 |
|  | BB | 0.55 | |
|  | LF | 0.16 | |
| X-TIME | XLM | 0.88 | **0.9** |
|  | MT | 0.84 | 0.88 |
|  | BB | 0.71 | |
|  | LF | 0.35 | |

Table 3: The results (Spearman correlations) of computing RILE via predicting all MARPOR/CMP sentence-level categories (CMP), RILE-specific categories (3-way), or using LITs (BB: BigBird; LF: Longformer).

### 4.1 Predicting MARPOR categories

As Table 2 shows, predicting the fine-grained MARPOR categories directly is a very hard task, both in the X-COUNTRY and X-TIME settings. Our models easily beat the majority-class baseline but only achieve an accuracy above 50% in the X-TIME setting with the XLM-ENC encoder.

Aggregating labels into the three RILE-relevant classes makes the task predictably simpler: the baseline F1 score rises from nearly zero to 0.44/0.49 (Other becomes the dominant category), but so does the performance of the models, to accuracies and F1 scores of 0.7 and above. However, there is still ample room for improvement. Interestingly, while using machine translation leads to consistent improvements in the X-COUNTRY setting, the X-TIME setting is better served with the multilingual encoder.[13]

---

[13]The results of using XLM-RoBERTa base in the X-TIME setting are as follows: CMP labels: accuracy – 0.51, F1 – 0.51, $r$ – 0.87; 3 labels: accuracy – 0.76, F1 – 0.75, $r$ – 0.88.

|          |     | X-COUNTRY   | X-TIME      |
|----------|-----|-------------|-------------|
| BigBird  | Acc | 0.69 / 0.73 | 0.74 / 0.71 |
|          | F1  | 0.68 / 0.71 | 0.72 / 0.68 |
| Longformer | Acc | 0.59 / 0.66 | 0.58 / 0.64 |
|          | F1  | 0.56 / 0.63 | 0.53 / 0.59 |

Table 4: Performance (on the chunk/manifesto level) of long-input Transformers on the task of 5-way political-stance classification. F1 scores are macro averaged and weighted by the frequency of the gold classes.

## 4.2 Computing RILE scores

**Label aggregation**  In agreement with our working hypothesis, Table 3 shows that even noisy labels can be used to calculate manifesto-wide scale values that are largely in agreement with gold values. When predicting RILE via label aggregation the best results are attained by using the multilingual encoder, both in the X-COUNTRY and in the X-TIME setting.

Somewhat surprisingly, aggregating the labels, even though this leads to a small number of surface-level classification mistakes, does not improve the eventual RILE scores in the X-COUNTRY setting ($r = 0.72$ from aggregated labels vs. 0.73 from all labels) and gives only a modest boost in the X-TIME setting (0.9 vs. 0.88).

**Long-input Transformers**  The performance of LITs is vastly uneven. In the X-COUNTRY setting, both models struggle: by-chunk RILEs from Longformer are essentially uncorrelated with gold ones, while BigBird's predictions show a non-negligible correlation (0.55), which is still much worse than the label aggregation results. In the X-TIME setting, while Longformer's predictions are still extremely noisy ($r = 0.35$), BigBird's ones are comparable to what the label aggregation approach achieves in the X-COUNTRY setting (0.71). As we discuss below, however, this correlation is somewhat misleading: while producing scores that are monotonically aligned with correct ones, BigBird predicts values that are very close to zero and thus differ greatly in their absolute values from the gold scores.

**LIT-based classifiers**  The results of the application of the better-performing LIT, BigBird, to the task of 5-way stance classification are shown in Table 4. Unlike RILE scores, by-chunk stance labels cannot be averaged, so for the final prediction each manifesto is assigned its majority class. The performance of the BigBird-based model in this setting is reasonable, with F1 scores ≈ 0.7.

|    | L | CL  | C   | CR | R |
|----|---|-----|-----|----|---|
| L  | **0** | 3   | 2   | 0  | 0 |
| CL | 0 | **133** | 135 | 1  | 0 |
| C  | 0 | 69  | **708** | 41 | 0 |
| CR | 0 | 0   | 70  | **28** | 0 |
| R  | 0 | 0   | 1   | 1  | **0** |

Table 5: Confusion matrix for the party stance predicted by the BigBird-based classifier in the X-COUNTRY setting. L: left, CL: centre left, C: centrist, CR: center right, R: right.

## 4.3 Error analysis

### 4.3.1 Regression to the mean

The distributions of gold RILE scores and those predicted in the X-COUNTRY setting by the best-performing label-aggregation pipeline and the best-performing LIT are shown in Figure 1.[14] The plots make it clear that both models are very conservative: predicted values cluster closer to the mean RILE score than in the gold data. BigBird is especially affected by this, which we take to indicate that it suffers from a lack of training data: the training dataset was big enough to correctly estimate the mean of the distribution but not big enough to approximate the correct dispersion.

The predictions of the label-aggregation model based on XLM-ENC approximate the dispersion much better. However, the model still fails to account for the heavy right tail in the gold data and presents a more symmetric picture. In terms of RILE scores, this corresponds to a *left skew*: the model often presents right-leaning manifestos (those with positive RILE scores) as more centrist.

A more detailed picture of the relationship between the gold RILE scores and those predicted by the label-aggregation model is shown, for both settings, in Figure 2, which also presents the density of the prediction errors. Consistently with Figure 1, the density of the X-COUNTRY error distribution has a slightly heavier left tail. To characterize this behavior, we can look at the cases where the sign of the prediction is flipped, i.e. the upper-left and the lower-right quadrants of the scatterplot. While the UL quadrant is nearly empty, the LR quadrant is populated not only near the $x = 0$ asymptote, but also further to the right. This suggests that in the cross-country and cross-lingual setting, the hardest aspect of the problem is correct identification of right-wing statements across countries.

[14]The situation in the X-TIME setting is similar. The corresponding plots are presented in Appendix D.

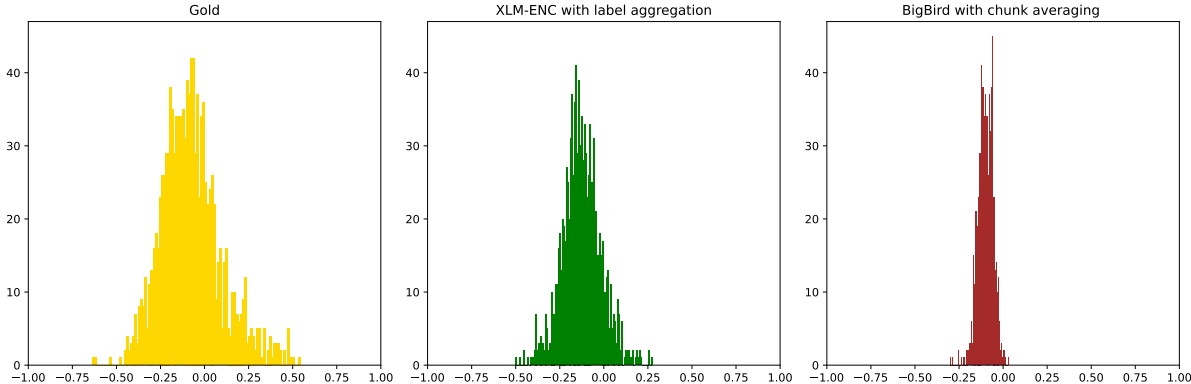

Figure 1: The distributions of gold and predicted RILE scores in the X-COUNTRY setting.

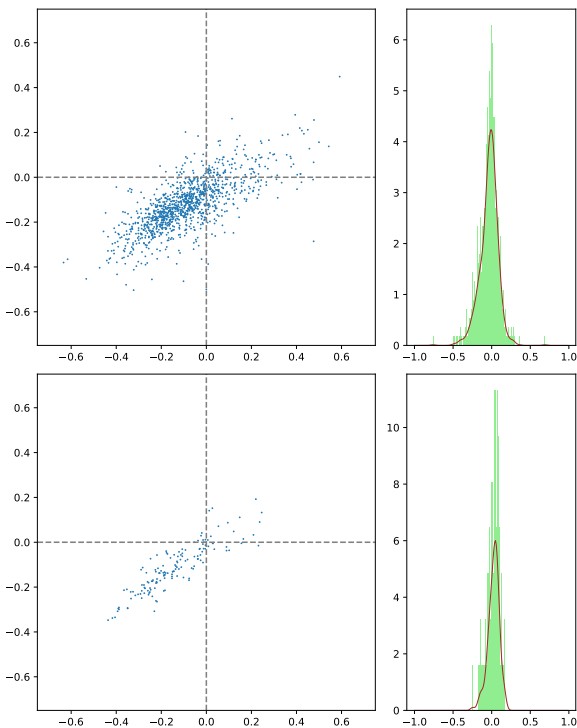

Figure 2: Gold vs. predicted RILE scores and histograms with density contours of the prediction errors in the X-COUNTRY (top) and X-TIME (bottom) settings with XLM-ENC and label aggregation.

One of the challenges associated with right-wing labels are their differing distributions across countries. While the variation in the cumulative share of left-wing labels in manifestos is bounded roughly between 0.2 and 0.3, with the same labels dominant everywhere, the variability of right-wing labels is much higher and their share is lower on average. See Figure 4 in Appendix E for details and Lachat (2018); Fielitz and Laloire (2021); Jahn (2022) for more in-depth analyses.

As the bottom panel of Figure 2 shows, the mag-

|        | Right | Left | Other |
|--------|-------|------|-------|
| Right  | **46** | 20   | 33    |
| Left   | 8     | **66** | 26    |
| Other  | 9     | 16   | **75** |

Table 6: Confusion matrix of coarse-grained labels used to compute the RILE score based on all MARPOR labels (the XLM-ENC + X-COUNTRY setting). True labels are in the rows, predicted labels in the columns.

nitude of errors in the X-TIME setting is considerably lower, with only a handful of sign-flip errors. This indicates that when a model has access to in-country data, the estimation of political positioning becomes easier, and the identification of right-wing tendencies is not a major hurdle any more.

The 5-way LIT-based party-stance classifier also suffers from the regression-to-the-mean problem, as can be seen in Table 5: the centrist category is overpredicted, while two extreme categories, which are rare in the data, are never predicted correctly.

### 4.3.2 Classifier errors and scaling analysis

One of the surprising results in Tables 2 and 3 is that low accuracy of the models trying to predict all MARPOR labels directly does not translate into low quality of respective RILE scores in the X-COUNTRY setting. This seems to suggest that errors of the models are not random: the models rather substitute, e.g., another Left-category label for a true Left-category label than replace a label from the Left set with a label from the Right set. A confusion matrix for the 3 coarse-grained labels (computed based on the *fine-grained* labels predicted by XLM-ENC in the X-COUNTRY setting) shown in Table 6 demonstrates that this is indeed the case.

## 5 Discussion

Our results show that multi-lingual automatic analysis of political-party positioning is at least partially feasible. It is possible to provide a high-level overview of the party system in a new country with a reasonable degree of precision, and even better results can be achieved with some amount of in-country data: the RILE scores computed using our method demonstrate a remarkably high correlation with the gold scores. Interestingly, the main obstacle to the success of our method seems to be not the language barrier, which is bridged well by either the off-the-shelf MT systems or the multilingual encoder, but the differences in the political culture across countries: the models struggle to correctly identify right-wing statements in the manifestos.

In practical terms, using long-input Transformers instead of sentence-level classifiers offers a way to greatly simplify the analysis and obviate the problems of subsentence identification in the input, as such models are able to make holistic judgements about long spans of text. In terms of performance, LITs struggle on the task of directly estimating RILE, compared to label-aggregation models, with the best model only approaching a reasonable level of performance. However, this must be taken with a grain of salt, since the label aggregation models have the advantage of gold-statement boundaries. Furthermore, our binned-regression experiment shows that LITs are promising candidates for coarse-grained party positioning analysis in terms of political 'camps'. For all models, the tails of the distribution remain hard to identify, with extreme categories rarely predicted correctly and centre left/centre right labels often mistaken for centrist.

## 6 Related work

The work on computational analysis of political documents traditionally employs bag-of-words methods, such as those popularised by Laver et al. (2003) and Slapin and Proksch (2008). Glavaš et al. (2017) introduce distributional semantics in the left–right analysis by using multilingual word alignment in the embedding space and a graph-based score-propagation algorithm. This approach is then built upon by Nanni et al. (2022).

Rheault and Cochrane (2020) adapt the word2vec methodology to the analysis of parliamentary speeches in a single-language setting via the use of trained party vectors, whose dimension-

ality they reduce using PCA; they then interpret one of the resulting axes as the left–right scale. Vafa et al. (2020) instead develop a methodology for identifying the political position of lawmakers on the progressive-to-moderate dimension with a bag-of-words-based topic-modelling approach.

The use of contextualised embeddings for political analysis has not yet become mainstream. Abercrombie et al. (2019) test a wide range of methods, from unigram statistics to BERT-based classifiers, for assigning MARPOR labels to classify debate motions from the UK parliament. Dayanik et al. (2022) use several pre-trained single-language BERT models for the task of political-statement classification in five languages. Facing the same issues of label-frequency imbalance and rare labels, they mitigate them to some degree by using the hierarchical organisation of MARPOR labels; they do not try to compute RILE scores.

Ceron et al. (2022) introduce sentence transformers (Reimers and Gurevych, 2019) into the problem space and fine-tune the embedding model itself in order to learn a politically informative distance measure between manifesto texts. Ceron et al. (2023) further extend this method to analyse inter-party differences with regard to major policy domains, such as Law and Order or Sustainability and Agriculture.

More generally, our work falls into the domain of zero-shot classification with test data coming from a country or a time period not covered by the training data. The question of whether machine translation (Schäfer et al., 2022) or multilingual encoders (Litschko et al., 2022) is better suited for cross-lingual transfer is still actively debated, and we explore both options. From another perspective, the task of identifying and characterising political positions from textual data abuts larger fields of stance detection and argument mining (Küçük and Can, 2020; Reimers et al., 2019).

## 7 Conclusion

In this paper, we have proposed the first series models that generalise the task of political-party positioning across countries and election cycles. We showed that the main challenge – predicting MARPOR labels across countries and election cycles with high accuracy – is, surprisingly, not a real barrier on the way to a highly precise multilingual scaling analysis. We experimented with the Standard Right–Left Scale (RILE score), which

is widely discussed in the political-science literature, and demonstrated that party manifestos can be effectively characterized in these terms using state-of-the-art multilingual modeling techniques applied to sentence-level classification with subsequent label aggregation and that even better results can be achieved via task-specific label clustering.

We further experimented with replacing the label-aggregation approach with long-input Transformers – both using regression and classification formulations – in order to obviate the task of identifying spans of statements from manifestos. These models demonstrate promising performance but still underperform the more traditional pipeline mimicking manual analysis.

Bridging the gap between long-input models and political analysis is an important avenue for future work, together with tackling other political dimensions and further widening the scope of the analysis.

## Limitations

The main limitations of our work are twofold, and both stem from our dependence on the categories and annotations produced by the MARPOR project:

1. The RILE scale that we target is computed based on the MARPOR category labels, and we do not test if our methodology can be easily projected to other categorisation schemes. However, given the important role of the MARPOR codebook in the political-science literature and the amount of annotated data already available, we hope that our work makes a valuable contribution to the debate.

2. In label-aggregation pipeline, we are dependent not only on the labels themselves but also on the way they are applied to manifestos: following previous work (Dayanik et al., 2022; Ceron et al., 2022), we use the sub-sentence boundaries selected by MARPOR annotators in order to assign a single category to each statement. In the manifesto texts, sentences therefore sometimes can be associated with several labels. There are several possible ways to address this issue (e.g., selecting a 'majority' label for each sentence in the training data, training a multi-label classifier, or learning splits together with labels from the training set), and they need to be explored to obtain best possible performance in real-world settings. Using LITs removes this issue, but their performance is not competitive.

## Acknowledgments

We acknowledge partial support by Deutsche Forschungsgemeinschaft (DFG) for project MARDY 2 (375875969) within priority program RATIO.

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

## A   Examples of the MARPOR categories

See Table 7.

## B   Names of MARPOR categories referenced by a number in the text

103 Anti-Imperialism

104 Military: Positive

105 Military: Negative

106 Peace

107 Internationalism: Positive

201 Freedom and Human Rights

201.1 Freedom

201.2 Human Rights

202 Democracy General

202.1 Democracy General: Positive

203 Constitutionalism: Positive

305 Political Authority

401 Free Market Economy

402 Incentives: Positive

403 Market Regulation

404 Economic Planning

406 Protectionism: Positive

406.1 Anti-Growth Economy:Positive

407 Protectionism: Negative

412 Controlled Economy

413 Nationalisation

414 Economic Orthodoxy

416 Anti-Growth Economy: Positive

501 Environmental Protection

502 Culture: Positive

504 Welfare State Expansion

505 Welfare State Limitation

506 Education Expansion

601 National Way of Life: Positive

602 National Way of Life: Negative

| Party | Text | Category |
|---|---|---|
| AfD | The principles of equality before the law. | Equality: Positive |
| CDU | We are explicitly committed to NATO's 2% target. | Military: Positive |
| FDP | And with a state that is strong because it acts lean and modern instead of complacent, old-fashioned and sluggish. | Governm. and Admin. Efficiency |
| SPD | There need to be alternatives to the big platforms - with real opportunities for local suppliers. | Market Regulation |
| Grüne | We will ensure that storage and shipments are strictly monitored. | Law and Order: Positive |
| Die Linke | Blocking periods and sanctions are abolished without exception. | Labour groups: Positive |

Table 7: Translated examples of sentences from German federal election manifestos (2021) with their categories as annotated by the Comparative Manifesto Project.

603 Traditional Morality: Positive

604 Traditional Morality: Negative

605 Law and Order

605.1 Law and Order: Positive

606 Civic Mindedness: Positive

607 Multiculturalism: Positive

608 Multiculturalism: Negative

701 Labour Groups: Positive

705 Unprivileged Minority Groups

706 Non-economic Demographic Groups

## C Dataset breakdown by country and by language

See Tables 8 and 9.

## D Distributions of predicted RILEs in the X-TIME setting

See Figure 3.

## E Cumulative share of left and right categories across countries

See Figure 4.

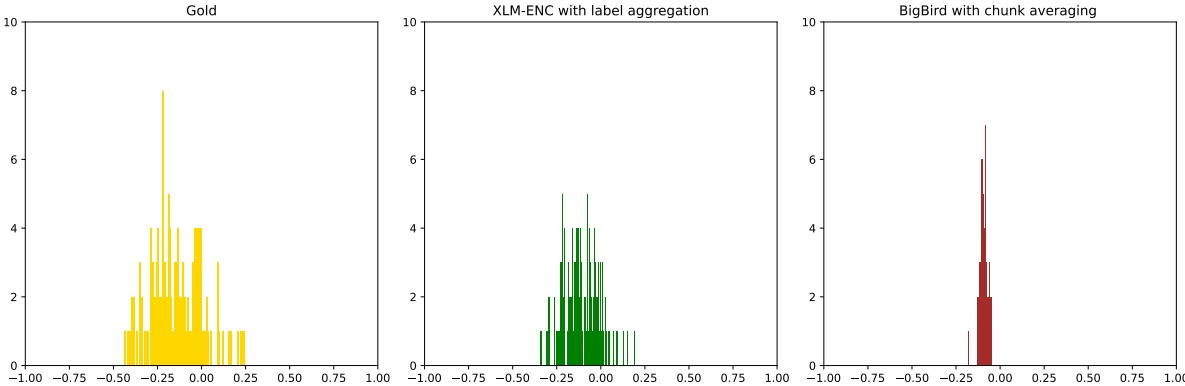

Figure 3: The distributions of gold and predicted RILE scores in the X-TIME setting.

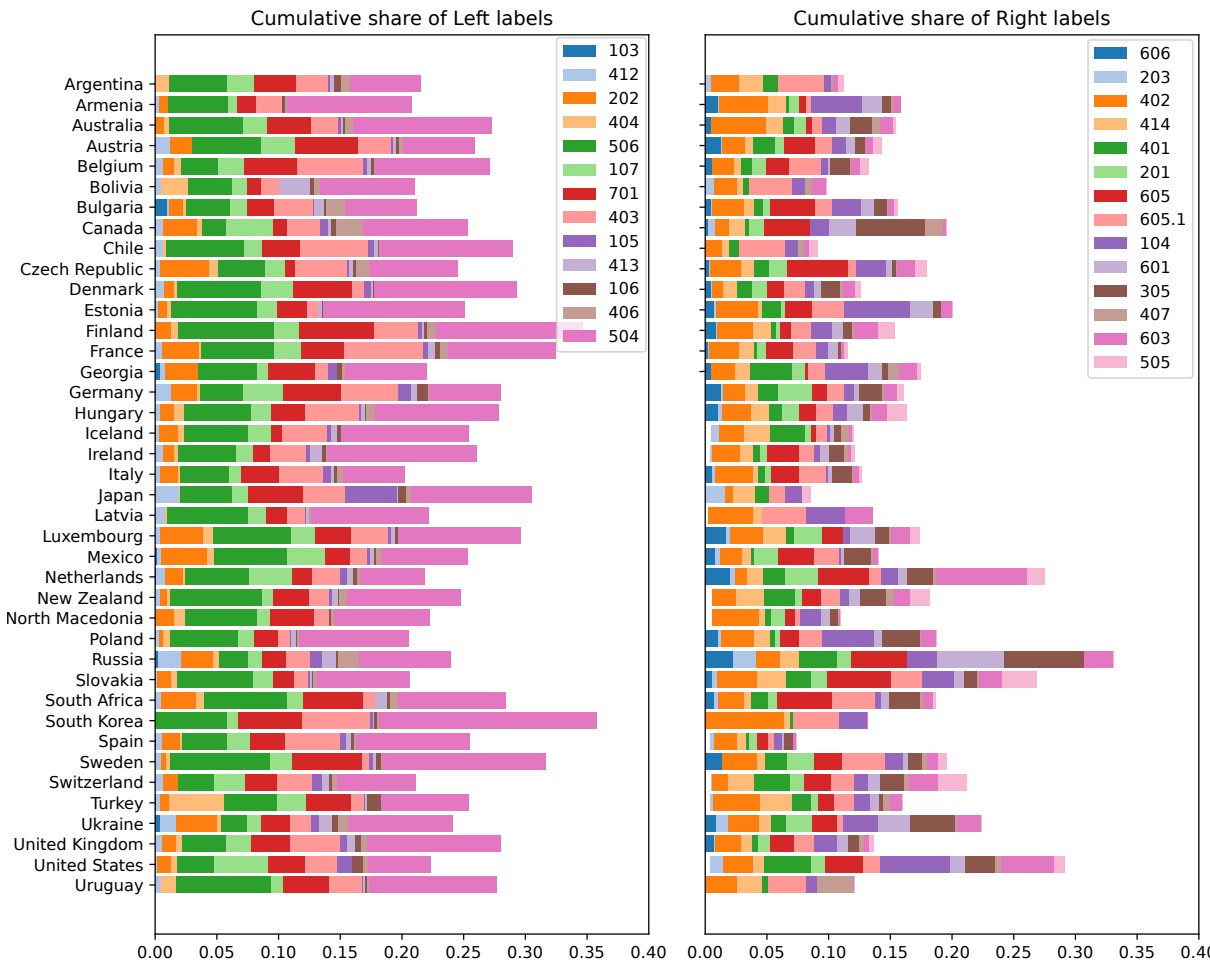

Figure 4: Cumulative shares of left-wing and right-wing labels in manifestos from different countries. See Appendix B for the explanation of label codes.

| Country | # manifestos | # sentences |
| --- | --- | --- |
| Argentina | 29 | 10983 |
| Armenia | 22 | 1623 |
| Australia | 30 | 21683 |
| Austria | 32 | 39452 |
| Belgium | 43 | 154699 |
| Bolivia | 9 | 7718 |
| Bulgaria | 18 | 8945 |
| Canada | 23 | 28524 |
| Chile | 17 | 33988 |
| Czech Republic | 31 | 25986 |
| Denmark | 45 | 17073 |
| Estonia | 23 | 16524 |
| Finland | 33 | 22520 |
| France | 20 | 9347 |
| Georgia | 19 | 2610 |
| Germany | 35 | 81759 |
| Hungary | 26 | 45246 |
| Iceland | 34 | 8139 |
| Ireland | 23 | 30348 |
| Israel | 1 | 24 |
| Italy | 32 | 22091 |
| Japan | 9 | 3387 |
| Latvia | 30 | 2030 |
| Luxembourg | 17 | 30768 |
| Mexico | 48 | 49818 |
| Netherlands | 45 | 72610 |
| New Zealand | 44 | 43869 |
| North Macedonia | 43 | 56719 |
| Poland | 30 | 27285 |
| Russia | 4 | 1350 |
| Slovakia | 33 | 25325 |
| South Africa | 24 | 12835 |
| South Korea | 5 | 6030 |
| Spain | 90 | 142878 |
| Sweden | 31 | 17293 |
| Switzerland | 50 | 20975 |
| Turkey | 23 | 54472 |
| Ukraine | 35 | 3099 |
| United Kingdom | 32 | 33211 |
| United States | 9 | 16262 |
| Uruguay | 5 | 16518 |

Table 8: Number of manifestos and number of sentences per country.

| Language code | Language | # manifestos | # sentences | Countries |
|---|---|---|---|---|
| bg | Bulgarian | 18 | 8945 | Bulgaria |
| ca | Catalan | 18 | 32780 | Spain |
| cs | Czech | 31 | 25986 | Czech Republic |
| da | Danish | 45 | 17073 | Denmark |
| de | German | 123 | 163622 | Austria, Germany, Italy, Luxembourg, Switzerland |
| en | English | 177 | 171812 | Australia, Canada, Ireland, Israel, New Zealand, South Africa, United Kingdom, United States |
| es | Spanish | 174 | 223047 | Argentina, Bolivia, Chile, Mexico, Spain, Uruguay |
| et | Estonian | 23 | 16524 | Estonia |
| fi | Finnish | 29 | 21313 | Finland |
| fr | French | 52 | 105570 | Belgium, Canada, France, Luxembourg, Switzerland |
| gl | Galician | 6 | 6076 | Spain |
| hu | Hungarian | 26 | 45246 | Hungary |
| hy | Armenian | 22 | 1623 | Armenia |
| is | Icelandic | 34 | 8139 | Iceland |
| it | Italian | 33 | 21646 | Italy, Switzerland |
| ja | Japanese | 9 | 3387 | Japan |
| ka | Georgian | 19 | 2610 | Georgia |
| ko | Korean | 5 | 6030 | South Korea |
| lv | Latvian | 30 | 2030 | Latvia |
| mk | Macedonian | 43 | 56719 | North Macedonia |
| nl | Dutch | 75 | 155807 | Belgium, Netherlands |
| pl | Polish | 30 | 27285 | Poland |
| ru | Russian | 4 | 1350 | Russia |
| sk | Slovak | 33 | 25325 | Slovakia |
| sv | Swedish | 35 | 18500 | Finland, Sweden |
| tr | Turkish | 23 | 54472 | Turkey |
| uk | Ukrainian | 35 | 3099 | Ukraine |

Table 9: Number of manifestos and sentences per language and respective source countries.