# OpenReview forum: "Multilingual estimation of political-party positioning: From label aggregation to long-input Transformers"
_EMNLP/2023/Conference — EMNLP 2023 Main_

### Official Review · Reviewer_gyFa · 2023-08-01

**Typos Grammar Style And Presentation Improvements:** 239 evaluate it one - -> evaluate it on
**Soundness:** 4

**Excitement:**

4: Strong: This paper deepens the understanding of some phenomenon or lowers the barriers to an existing research direction.

**Paper Topic And Main Contributions:**

The paper describes NLP experiments for automatic scaling analysis (party positioning on the RILE scale proposed by the CMP, the Comparative Manifesto Project,) of party manifestos using label aggregation (Left/Right/Other) and long-input Transformer-based models which make the prediction from raw text. The experiments are run on 2 settings, leave-one-country-out and old-vs-new time.

**Questions For The Authors:**

Could you shortly discuss cases in which your approach can be useful?
Does the red contour in the RILE scores of figure 2 refer to the Gold or the predicted RILE scores?
Do you think that using automatically extracted sentences instead of the quasi-sentences used by the CPM would change the results significantly?

**Reasons To Accept:**

The study is interesting, and it gives some promising results using long-input Transformer models on the label aggregation, while annotating the quasi-sentences with the many labels manually annotated in the CPM is, not surprisingly,  difficult. All steps of the experiments are well described, and Spearman's correlation coefficient is used for relating their scores to the scores from the gold dataset, showing that on the cross-country and multi-lingual setting is difficult to individuate right-wing statements.

**Reasons To Reject:**

The utility of predicting  a party position as Left/Right/Other without considering on which specific policy issues the party has left/centre/right positions can be in a few cases interesting, however in many countries parties' orientation is known. I suggest the authors to motivate further their work. More interesting is to have a party’s positioning w.r.t. specific issues, which is the motivation behind the CPM’s manual annotations.

**Reproducibility:**

5: Could easily reproduce the results.

**Reviewer Confidence:**

4: Quite sure. I tried to check the important points carefully. It's unlikely, though conceivable, that I missed something that should affect my ratings.

---

> ### Author Rebuttal · Authors · 2023-08-28
>
> We thank the reviewer for the comments. We disagree that our problem statement is unimportant because “in many countries parties' orientation is known”, for at least two reasons. First, this information may be inaccessible in other countries or across longer time spans. Second, pre-theoretical assessments of party positions are typically specific to individual countries, but party comparison across countries is a relevant field of investigation that ideally requires a country-agnostic operationalisation of such scales. The aim of the MARPOR project is to make this kind of information universally available, but – as we mention in the paper – its coverage is incomplete both in terms of countries and election cycles. This motivates our use of NLP for generalised analysis.
>
> As for capturing of parties’ stances on particular topics, this is an interesting research question that is largely orthogonal to our focus on left–right scaling, which, we would like to stress, is accorded a lot of value by the political-science community (cf. https://doi.org/10.1177/13540688211026076). There is recent work that tries to compare parties based on their policy-domain-specific views (e.g. Ceron et al. ACL Findings 2023), and we consider it complementary to ours as it does not provide whole-manifesto RILE estimates.
>
> “Could you shortly discuss cases in which your approach can be useful?”
>
> Examples include the analysis of short-term policy and public-opinion shifts between election cycles (where we reduce the time lag between the release of the manifestos and their analysis) and the analysis of the influence of extraneous factors, such as economics, climate change, or large-scale international events, on the political landscape.
>
> “Does the red contour in the RILE scores of figure 2 refer to the Gold or the predicted RILE scores?”
>
> The red contour is the density estimate corresponding to the histograms of prediction errors in two different experimental settings. We will make it clearer in the caption.
>
> “Do you think that using automatically extracted sentences instead of the quasi-sentences used by the CPM would change the results significantly?”
>
> Our general guess would be that it would not because there is no evident reason why any of the strategies for dealing with raw sentences (only using the first-statement label or the majority label, or learning the splits) should bias the method in either direction, such that the eventual RILE predictions become skewed. However, to answer this question rigorously we need to make estimates vis-à-vis different strategies.

---

### Official Review · Reviewer_VGTx · 2023-08-04

**Soundness:** 4

**Excitement:**

4: Strong: This paper deepens the understanding of some phenomenon or lowers the barriers to an existing research direction.

**Paper Topic And Main Contributions:**

The paper presents a novel approach for classifying party manifestos based on the MARPOR framework and RILE categories. Two different approaches are used, a label aggregation approach that makes the final assessment based on the aggregation of statement-level labels, and a long-input transformer approach, that provides a direct classification based on the whole manifesto.

**Reasons To Accept:**

* The paper works with an interesting and relevant dataset and compares different approaches and technologies
* The experimental setup is well described and the paper presents an insightful analysis of the errors that occurred
* The paper can provide a starting point for interesting discussions and future work

**Reasons To Reject:**

I do believe that from a social science perspective, the practical relevance of the results is rather low because of two main shortcomings:
* The approach was only reliable in generating manifesto-level classifications on the more high-level RILE-scale. The approach does not provide insights into the specific stances of a party in different areas or an explanation for the final assessment. Therefore, I would assume that social scientists would benefit relatively little from this very high-level classification, partially also because the approach does not provide any explanations for the assessment and scientists would therefore would be unable to directly assess the reliability of the assessment without reading the whole manifesto.
* As the analysis of the results shows, the performance of the classifier is particularly weak at the edges of the left-right-spectrum, because it tends towards the middle. Arguably, the area around the edges is particularly interesting for many political analyses.

That said, I do believe these (at least partially) negative results are valuable because the authors present such a clear analysis of the errors and the task at hand is difficult and the paper can provide a great starting point for future work and can spark a discussion about the task in the EMNLP community.

**Reproducibility:**

4: Could mostly reproduce the results, but there may be some variation because of sample variance or minor variations in their interpretation of the protocol or method.

**Reviewer Confidence:**

3: Pretty sure, but there's a chance I missed something. Although I have a good feel for this area in general, I did not carefully check the paper's details, e.g., the math, experimental design, or novelty.

---

> ### Author Rebuttal · Authors · 2023-08-28
>
> We thank the reviewer for the comments and we largely concur with the highlighted limitations of our work. However, in our point of view, they partly stem from the problem statement, which we still find practically useful.
>
> Our method provides generally reliable, if not very fine-grained, estimates of dominant political tendencies on the by-party and by-election basis, which is relevant for the analysis of short-term policy and public-opinion shifts between election cycles or the study of the influence of extraneous factors, such as economics, climate change, or large-scale international events, on the political landscape. We regard it as an open question whether social scientists will use the proposed approach given that they previously have not had access to this kind of tool, and will endeavour to publicise the tool in that community.
>
> As for capturing of parties’ stances on particular topics, this is an important area of research, and there is recent work (such as Ceron et al. ACL Findings 2023) that tries to tackle it directly without providing whole-manifesto RILE estimates. However, the ostensibly simplistic question of analysing political texts in terms of left—right scaling has proven to be of immense interest to political scientists (cf. the overview in the monograph Left and Right: The Small World of Political Ideas by Christopher Cochrane), and we are confident that they will welcome computational developments in this area.
>
> Finally, regarding the fact that our model often fails to predict radical tendencies in manifestos, we agree that this is a limitation, but the main reason for it is the fact that texts exhibiting such tendencies are very infrequent. Investigating methods to improve their identification for the benefit of the scholars who study extremism is an interesting area for future work. We will add this to the draft.

---

### Official Review · Reviewer_MK6B · 2023-08-04

**Soundness:** 4

**Excitement:**

4: Strong: This paper deepens the understanding of some phenomenon or lowers the barriers to an existing research direction.

**Missing References:**

Previous work on transformer-based prediction of policy preferences using the categories from the Manifesto project (these should be the same as your MARPOR categories): https://aclanthology.org/K19-1024.pdf

**Paper Topic And Main Contributions:**

This paper explores various methods for estimating RILE scores in a multilingual setting, including aggregation of predicted MARPOR category labels and direct estimation using transformer-based regression models. The methods are evaluated experimentally in both a diachronic and "country-centric" setting.

**Questions For The Authors:**

I would appreciate if you could briefly answer my question under "Reasons To Reject", thanks!

**Reasons To Accept:**

- (+) Overall solid work using state-of-the-art models for a central task in CSS / poltext
- (+) Novel approach when compared with previous approaches (i.e., multilingual RILE estimation as opposed to multilingual scaling)
- (+) Thorough evaluation

**Reasons To Reject:**

- (-) Limited/missing comparison with previous work. My only problem with this paper is its limited experimental comparison with previous work. The authors mention a few previous multilingual approaches for multilingual political text scaling (Glavas et al. 2017; Nanni et al., 2022; Rheault and Cochrane 2020): how good are your results in relation to theirs? I understand that their problem setting is slightly different than yours (e.g., vanilla scaling vs. predicting RILE, different evaluation measures), but at the very minimum, there should be a discussion of why you cannot compare (i.e., are RILE scores not comparable with scaling ones? If yes, why?).

**Reproducibility:**

5: Could easily reproduce the results.

**Reviewer Confidence:**

5: Positive that my evaluation is correct. I read the paper very carefully and I am very familiar with related work.

---

> ### Author Rebuttal · Authors · 2023-08-28
>
> We thank the reviewer for the thoughtful comments.
>
> The exact meaning of “scaling” varies widely across studies. E.g., Rheault and Cochrane (2020) project party embeddings learned using word2vec from speeches from British, American, and Canadian legislative assemblies on two principal components and then discuss the resulting 2D patterns. They interpret one of the resulting dimensions as the left–right scale and report its correlation with RILE values assigned to respective parties; the correlations are 0.63 (US), 0.68 (UK), and 0.77 (Canada). These numbers are incomparable to ours because R&C are not trying to assign any numbers to manifestos, while RILE values are not defined for parliamentary speeches. Glavas et al. and Nanni et al. also target parliamentary speeches, with both studies using different projection axes/scales.
>
> The biggest methodological overlap with our work is that by Dayanik et al. (2022), who assign categorical labels to political statements from manifestos. However, there are several differences in setup: they do not compute RILE values based on this analysis; they cover texts in only 5 languages; they randomly split train/test data within languages (i.e. do X-time but not exactly) and use five different single-language BERT models instead of a multilingual setting. Thus, it is possible to compare the performance of our label classifiers (F1=0.55 X-time) with their models (F1 between 0.35 and 0.45), but it is unclear whether the comparison makes sense.
>
> In sum, we believe that a quantitative comparison of our results to prior work is not well defined, and consequently we did not address it in the draft. We will be happy to provide more background detail in the revised version.

---

### Meta-Review · Area_Chair_Ajjm · 2023-09-11

**Recommendation:** 5

**Metareview:**

This paper tackles two tasks: predicting the 143 different categories of the ManIfesto Research on Political Representation project (MARPOR) and RILE values, Standard Right–Left Scale that uses pre-defined right-wing and left-wing categories from the MARPOR project and a deterministic formula to get a RILE score between -1 and 1. They highlight that there needs to be more methods that deal with long texts and work robustly across domains and languages.

Reviewers generally found the experimental set-up to be very sound.

Reviewers had a few questions about the motivations and impact of the substantive task (the predicting MARPOR categories and RILE scores). However, the authors wrote a pretty convincing rebuttal that the tasks are “relevant for the analysis of short-term policy and public-opinion shifts between election cycles or the study of the influence of extraneous factors, such as economics, climate change, or large-scale international events, on the political landscape” and “pre-theoretical assessments of party positions are typically specific to individual countries, but party comparison across countries is a relevant field of investigation that ideally requires a country-agnostic operationalisation of such scales.” These do seem to be well-established tasks that social scientists would benefit from having improved predictive accuracy.

Overall, it seems this is a sound and exciting paper that spans *both* the NLP and social science communities. I think both communities could potentially benefit from this paper.

---

### Decision · Program_Chairs · 2023-10-07

**Decision:**

Accept-Main

**Comment:**

This paper tackles two tasks: predicting the 143 different categories of the ManIfesto Research on Political Representation project (MARPOR) and RILE values, Standard Right–Left Scale that uses pre-defined right-wing and left-wing categories from the MARPOR project and a deterministic formula to get a RILE score between -1 and 1. They highlight that there needs to be more methods that deal with long texts and work robustly across domains and languages.

Reviewers generally found the experimental set-up to be very sound.

Reviewers had a few questions about the motivations and impact of the substantive task (the predicting MARPOR categories and RILE scores). However, the authors wrote a pretty convincing rebuttal that the tasks are “relevant for the analysis of short-term policy and public-opinion shifts between election cycles or the study of the influence of extraneous factors, such as economics, climate change, or large-scale international events, on the political landscape” and “pre-theoretical assessments of party positions are typically specific to individual countries, but party comparison across countries is a relevant field of investigation that ideally requires a country-agnostic operationalisation of such scales.” These do seem to be well-established tasks that social scientists would benefit from having improved predictive accuracy.

Overall, it seems this is a sound and exciting paper that spans *both* the NLP and social science communities. I think both communities could potentially benefit from this paper.